# Modulation of Intracellular Copper Levels as the Mechanism of Action of Anticancer Copper Complexes: Clinical Relevance

**DOI:** 10.3390/biomedicines9080852

**Published:** 2021-07-21

**Authors:** Maria V. Babak, Dohyun Ahn

**Affiliations:** Drug Discovery Lab, Department of Chemistry, City University of Hong Kong, 83 Tat Chee Avenue, Hong Kong 999077, China; dohyunahn2-c@my.cityu.edu.hk

**Keywords:** Cu homeostasis, cancer, Cu overload, Cu chelators, Cu ionophores, Cu toxicity, chemical biology, thiosemicarbazones, clinical trials, immune system

## Abstract

Copper (Cu) is a vital element required for cellular growth and development; however, even slight changes in its homeostasis might lead to severe toxicity and deleterious medical conditions. Cancer patients are typically associated with higher Cu content in serum and tumor tissues, indicating increased demand of cancer cells for this micronutrient. Cu is known to readily cycle between the +1 and +2 oxidation state in biological systems. The mechanism of action of Cu complexes is typically based on their redox activity and induction of reactive oxygen species (ROS), leading to deadly oxidative stress. However, there are a number of other biomolecular mechanisms beyond ROS generation that contribute to the activity of anticancer Cu drug candidates. In this review, we discuss how interfering with intracellular Cu balance via either diet modification or addition of inorganic Cu supplements or Cu-modulating compounds affects tumor development, progression, and sensitivity to treatment modalities. We aim to provide the rationale for the use of Cu-depleting and Cu-overloading conditions to generate the best possible patient outcome with minimal toxicity. We also discuss the advantages of the use of pre-formed Cu complexes, such as Cu-(bis)thiosemicarbazones or Cu-*N*-heterocyclic thiosemicarbazones, in comparison with the in situ formed Cu complexes with metal-binding ligands. In this review, we summarize available clinical and mechanistic data on clinically relevant anticancer drug candidates, including Cu supplements, Cu chelators, Cu ionophores, and Cu complexes.

## 1. Briefly about the Physiological Control of Cu Balance 

Cu is an essential endogenous metal involved in various biological functions, including but not limited to the regulation of energy production and other redox reactions, as well as regulation of glucose, cholesterol, and iron metabolism [1]. Medicinal value of Cu was recognized many centuries ago: in Hippocrates’ time Cu mixtures were used for treatment of pulmonological and mental disorders [2], and Ayurvedic medicine recommended to store drinking water in Cu vessels due to their antibacterial properties [3]. However, Cu is potentially a highly toxic metal and its imbalance (both overload and depletion) is associated with severe health disorders, therefore, its balance is tightly controlled [1,4]. Upon absorption into the bloodstream, Cu(II) rapidly binds to plasma proteins, such as ceruloplasmin (≈70% of Cu), transcuprein, and albumin, which transport Cu^2+^ ions to plasma membrane. Subsequently, it undergoes reduction to Cu(I) by cupric reductases and only after reduction Cu(I) ions are transported into the cells via Cu transporter 1 (Ctr1) and Cu transporter 2 (Ctr2). Following internalization of Cu(I) ions, they are delivered to Cu-requiring proteins by Cu chaperons, namely cytochrome c oxidase Cu chaperone 17 (COX17), Cu chaperone for Cu/Zn superoxide dismutase, and antioxidant protein 1 (Atox1). The major Cu-requiring proteins are cytochrome C oxidase (COX), Cu/Zn superoxide dismutase, and ATPase Copper Transporting Alpha and Beta (ATP7A and ATP7B). ATP7B is also responsible for excretion of Cu through the bile duct. For more detailed information please refer to the following reviews [1,5,6,7,8,9,10,11,12,13].

Since an excess of unbound Cu is highly toxic to the body, the concentrations of Cu ions in cytoplasm are maintained at exceptionally low levels, in fact, less than 1 free Cu ion per cell (10^−15^–10^−21^ M) [14]. This cytoplasmic Cu control is mainly achieved through its binding to glutathione (GSH), a tripeptide with powerful antioxidant function. GSH is present in cytoplasm in a millimolar range, which greatly exceeds the concentration of free Cu ions, therefore, GSH serves as a function of cytosolic Cu buffer to prevent high concentrations of unbound toxic Cu ions and to establish a negative metal gradient for Ctr1-regulated Cu transport [10]. In addition, excess of metal ions is sequestered by metallothionines (MTs), which are transcriptionally activated by high concentrations of toxic metal ions, including Cu. MTs are rich in cysteines and therefore can irreversibly form stable Cu-thiolate clusters, thus protecting cells from Cu overload. It is believed that MTs and GSH create a challenge for Cu drug design, since various existing redox-active Cu complexes are unstable towards the MT/GSH system and might not be efficiently delivered to their proposed biomolecular targets in cytosol and nucleus.

## 2. Systemic Changes in Cu Homeostasis of Cancer Patients

Since copper plays an important role in the cellular function and metabolism, the question arises as to whether there are significant differences in Cu levels in cancer patients in comparison with healthy cohorts, and if yes, whether Cu levels can be used as a reliable biomarker of disease progression and response to treatment [15]. Cu concentrations were determined from emission spectrograms or by flame atomic absorption spectrometry in whole blood, plasma [16] or serum [15], toenail [17], hair [18], saliva [19], and tissues [20]. Many studies reported a positive correlation between circulating Cu levels and cancer progression, in particular breast cancer. A meta-analysis of 36 studies between 1991 and 2019 revealed significantly increased serum Cu levels in breast cancer patients in comparison with benign breast diseases and healthy controls [21]. Similar correlations have been observed in meta-analyses of other cancer types, including lung [22,23], colorectal [24], hepatocellular carcinoma [25], and other cancers, as well as in animal models [26]. It was shown that hepatocellular carcinoma tissues in mice were characterized by approximately 10-fold higher Cu levels than healthy liver tissues [26]. In contrast, a number of meta-analyses did not observe any statistically significant correlations, which might be related to the differences in number of subjects, genetic background of patients, sample type, etc. [27,28,29]. Generally, there is a common trend of increased Cu accumulation in serum and tumor tissues of cancer patients; however, the data about association between Cu levels and increased cancer risk is inconclusive. While several studies revealed a possible link between Cu levels and increased risk of colorectal cancers [24], it was not observed for hepatocellular carcinoma [30] and lung cancer [31]. Similar to Cu levels, the ratio of serum Cu/Zn levels was also significantly higher in cancer patients compared to healthy controls, which typically resulted from the increase of Cu levels with simultaneous decrease of Zn levels. Recently, several studies aimed to determine whether circulating Cu levels in serum and full blood, and the ratio of Cu/Zn levels can serve as predictors of the patients’ mortality [32,33,34,35]. It was demonstrated that higher serum Cu levels, as well as Cu/Zn ratio, were associated with increased all-cause mortality in lung cancer patients [32]. This might be associated with the increase of Cu levels and Cu/Zn ratio at advanced stages of the disease, as well as during metastatic progression, thus resulting in lowered survivability [32,33,34,35]. It was suggested that elevation of serum/blood Cu levels might be a consequence of Cu ions release from necrotic tumor tissues [36] or a result of an inflammatory condition [37,38]. However, the increase of both circulating and intra-tumoral Cu levels suggests the systemic interference with Cu homeostasis in cancer patients.

To determine whether Cu levels might also correlate with patients’ response to anticancer treatment, Cu serum concentrations were determined in patients with hematologic malignancies before and after systemic treatment. It was revealed that elevated Cu levels significantly decreased to almost normal levels when patients went into remission and returned to pre-therapy values upon relapse [39]. Similarly, when serum Cu levels were monitored in patients with solid malignancies undergoing radiotherapy, the highest decrease was observed in patients with complete clinical response [40,41], indicating that Cu levels might indeed serve as a potential biomarker. The levels of ceruloplasmin were also shown to rapidly decrease upon tumor removal [42,43]. It should be noted that when Cu levels were measured in breast cancer patients before chemotherapy (Adriamycin) and after 3 courses of chemotherapy, no changes in Cu levels were detected [44]. 

Since both Cu and Zn consist of several naturally occurring isotopes (Cu: ^65^Cu (69.17%) and ^63^Cu (30.83%); Zn: ^64^Zn (48.6%), ^66^Zn (27.9%), ^67^Zn (4.1%), ^68^Zn (18.8%), and ^70^Zn (0.6%)), each biological system is characterized by specific isotopic fingerprints. It was suggested that disease-associated changes in such fingerprints might be more reliable predictive biomarkers than changes in Cu and Zn concentrations [45,46]. It was shown that ^65^Cu/^63^Cu ratio in serum of breast and colon cancer patients was significantly lower than in healthy individuals; moreover, the drop in ^65^Cu isotope in serum occurred several months earlier than the changes in other potential biomarkers were observed [46]. Similarly, the blood and tumors of hepatocellular carcinoma patients were ^65^Cu-depleted and ^65^Cu-enriched, respectively [47]. It was suggested that heavy ^65^Cu isotope was preferentially chelated in cancer cells with the subsequent release of lighter ^63^Cu into bloodstream [46,47]. However, analysis of ^65^Cu in patients with hematological malignancies revealed similar decrease of heavier Cu isotope in blood of cancer patients in comparison with healthy controls [48]. The changes in Cu isotopic pattern in cancer patients were also linked to Warburg effect. It was suggested that more isotopically heavy Cu^2+^ was efficiently chelated by L-lactate, while isotopically light Cu^+^ was excreted into bloodstream [46]. Therefore, increased lactate production in cancer cells resulted in increased chelation of ^65^Cu isotope and increase of ^65^Cu/^63^Cu ratio in cancer tissues and decrease of ^65^Cu/^63^Cu ratio in serum. Another possible explanation of low ^65^Cu isotope was its release from endogenous Cu stores, such as MTs [47].

In contrast to Cu, the differences in Zn isotopic distribution in cancer patients and healthy controls were less informative. It was shown that blood plasma of cancer patients with hematologic malignancies was characterized by higher abundance of heavier isotopes than in healthy people [48]. However, resections of breast tumors revealed lighter Zn isotopic content than healthy breast tissues, while serum of breast and colon cancer patients and healthy individuals could not be differentiated [46,49,50]. 

## 3. Tumor Modulation with Inorganic Cu Salts

### 3.1. Carcinogenesis

In light of higher Cu demand in tumors, it appears reasonable to question whether dietary Cu supply might affect the proliferation of cancer and healthy cells or whether Cu can be even considered a carcinogen. In animals, long-term Cu supplementation resulted in significant toxicity and apoptosis but no relationship between Cu and cancer progression has been found. When rats [51,52,53,54] or mice [55,56,57] were given CuSO_4_, CuCl_2_, or CuCO_3_ for up to 2 years, significant apoptosis in liver [51,56,58], lungs [55], brain [52,53,58], and kidneys [58] has been observed, which was associated with ROS production, reduced glutathione levels, lipid peroxidation injury, and mitochondrial swelling. Nevertheless, chronic Cu exposure was not associated with the development of neoplasms [57,59], thus indicating low carcinogenic potential of Cu. 

Several studies have investigated the effect of Cu supply on cancer incidence and progression in the presence of carcinogens. One of the first studies investigating the role of Cu substances in the mouse carcinogenesis was reported in 1914 [60]; however, no conclusive results were obtained. While inorganic Cu salts did not show any effects on the tumor growth, some colloidal Cu formulations markedly slowed down the tumor progression depending on the type of formulation and initial size of the tumors [60]. Subsequently, rats with high-Cu diet (food or water supplemented with 0.15–0.5% of CuSO_4_ or Cu(OAc)_2_) and normal diet were treated with dimethylaminoazobenzene (DMAB), its analogues, or other carcinogens, and the tumor incidence and progression were monitored [61,62,63,64,65]. It was found that rats fed with carcinogens and Cu salts were characterized by the significant delay in liver tumor progression [63] or significantly decreased tumor incidence [59,61,64,66], possibly due to the decreased cellular intake of carcinogens in the presence of Cu or competitive binding for the available protein sites in the liver [62,65,67,68]. The supplementation of Cu-rich diet correlated with increased Cu content in rat livers [59,61,62,67], in particular mitochondria [66]. Although some concerns were raised that reduction of tumor incidence might be related with the Cu-induced destruction of carcinogenic azo-dyes [65], but not other types of dyes [69], some studies demonstrated high stability of tested carcinogens in the presence of Cu [61].

In contrast, supplementation of water of pseudopregnant mice with CuSO_4_ (198 mg/L) did not show any effect on the incidence of carcinogen-induced lymphomas, lung, ovaries, and breast cancers, which was most probably related to insufficient Cu concentrations in tissues (were not measured) [70]. Moreover, several studies demonstrated that supplementation of food and water with Cu salts has stimulated tumor progression [57,71,72,73]. When rats were given food with CuSO_4_ (42.6 mg Cu/kg food) with or without antioxidant resveratrol, the carcinogen-induced mammary carcinogenesis was markedly accelerated [71,72]. This process was accompanied by the significant increase of serum Cu and decrease of serum Fe and Zn/Cu ratio similar to those observed in breast cancer patients [74,75]. In addition, the decreased expression of antioxidant enzymes, such as catalase, glutathione peroxidase, and superoxide dismutase, was observed [72], which play a protective role against ROS and oxidative stress. Since free radicals contribute to the promotion of various stages of carcinogenesis [76], reduced activities of antioxidant enzymes might be related to the observed effects in rats. It was also shown that Cu-rich diet aggravated carcinogen-induced genomic instability [71]. 

Hanahan et al. employed a genetically engineered mouse model of multistage pancreatic neuroendocrine tumor development and studied the effects of Cu in drinking water on the process of tumorigenesis (Figure 1) [57]. The mice were given 20 μM of CuSO_4_ for ≈11 weeks, which corresponds to the maximal allowed Cu levels in public water supplies [57]. It was demonstrated that tumors in mice, drinking Cu-enriched water, were at least 2-fold larger than in control group, indicating the role of Cu in tumorigenesis. Accordingly, when systemic levels of Cu were decreased using Cu chelating agent tetrathiomolybdate (TM), plasma Cu levels in mice decreased by 40%, corresponding to significant reduction of tumor growth. Importantly, the restriction of bioavailable Cu delayed the onset of angiogenic switch in pre-malignant lesions, thereby deferring further tumor development. Subsequent analysis of tumors demonstrated significant increase of proliferating cells in Cu-treated group and significant decrease of proliferating cells in mice treated with Cu chelator, indicating the role of Cu as a rate-limiting nutrient [57]. Similar effects have been observed in a mouse model of spontaneous lung adenocarcinoma, where tumorigenesis was accelerated by addition of CuSO_4_ to the drinking water and slowed down by addition of Cu chelators, respectively [73].

In addition to the relationship between high-Cu diet and cancer development, several studies investigated the effects of low dietary Cu intake on spontaneous carcinogenesis. When multiple intestinal neoplasia (Min) mouse pups and their nursing dams were fed with the low-Cu diet (CuCO_3_, 1 ppm Cu) [77] for 13 weeks, the animals demonstrated significantly higher incidence of tumors in small intestines than the control group (6 ppm Cu) [78]. In agreement, a low-Cu diet in rats was associated with higher formation of carcinogen-induced aberrant crypt foci (ACF), which are the earliest identifiable preneoplastic lesions of colon cancer [79], as well as spontaneous tumors in small intestines [80], than in rats with adequate diet. Low copper intake was associated with the decreased expression of Cu-containing enzymes, such as Cu/Zn superoxide dismutase and ceruloplasmin, leading to oxidative stress. Another possible explanation might be decreased expression of protein kinase C isozymes in animals with low Cu diet, which were reported to inhibit cancer cell growth and proliferation [81].

### 3.2. Proliferation

The effects of inorganic Cu salts on the proliferation and viability of various cell lines have been studied, including hepatic stellate cells (HSCs), hepatocytes, endothelial cells, hematopoietic progenitors, etc. HSCs are undifferentiated quiescent cells that reside in the liver; however, they play a significant role in the process of liver carcinogenesis and represent a major component of hepatocellular carcinoma tumor environment. While no effect has been observed when HSCs were treated with 50 μM of CuSO_4_, they demonstrated markedly accelerated growth when the concentration of CuSO_4_ was elevated to 200 μM [82]. The accelerated growth was accompanied by increased oxidative stress, reflected by high ROS production, and lipid peroxidation, as well as reduced GSH levels [82]. On the other hand, hepatocellular carcinoma cells HepG2 that were grown in cell culture media supplied with 20–160 μM CuSO_4_ demonstrated reduced proliferation and eventually underwent ROS-mediated apoptosis [83,84]. 

Similar to HSCs, the proliferation of human umbilical vein endothelial cells (HUVEC) has been stimulated with CuSO_4_ (500 μM). However, when human dermal fibroblasts and arterial smooth muscle cells were tested under identical conditions, no effects on proliferation rate and viability were observed [85]. In an attempt to decipher Cu-induced stimulation of cell proliferation, mouse embryonic fibroblasts (MEFs), as well as Atox1-null MEFs (where Atox1 is Antioxidant-1, one of the major Cu chaperones) were treated with 10 μM of CuCl_2_. The addition of Cu markedly stimulated proliferation of wild-type MEFs, but not Atox1-null MEFs, indicating that Cu-induced proliferation was Atox1-dependent [86]. It was shown that CuCl_2_ activated nuclear translocation of Atox1, DNA binding, and transcription activity, thereby accelerating cell proliferation [86]. Typically, highly proliferating tumors are characterized by selective Cu accumulation in the nuclei, whereas normal tissue predominantly accumulates Cu within the cytoplasm [87,88,89]. In agreement with this observation, cancer cells are characterized by increased nuclear translocation of Atox1, which correlated with the severity of the disease; therefore, nuclear status of Atox1 might determine the proliferation status of the cells [90,91]. 

### 3.3. Differentiation

Besides regulating cell proliferation, inorganic Cu salts were also reported to affect cell differentiation. The differentiation of hematopoietic stem/progenitor cells (HSPCs) was monitored in Cu-rich and Cu-deficient cell culture media, which was obtained by the addition of 10 μM CuSO_4_ or CuCl_2_ and Cu chelator, respectively [92,93,94]. It was shown that addition of Cu salts significantly impaired the proliferation of progenitor cells and accelerated differentiation. Accordingly, cells treated with a Cu chelator (tetraethylenepentamine) resulted in the decrease of chelatable Cu levels and has led to the enrichment of progenitor subsets, as well as increasing their long-term culture potential [92,93,95,96]. These observations have clinical relevance, since Cu-deficient patients with neutropenia (low neutrophil count) were characterized by the absence of mature cells in their bone marrow [97]. Similar results have been observed with mesenchymal stem cells (MSCs), which are undifferentiated cells that reside in bone marrow. When MSCs were treated with 50 μM of CuSO_4_, their proliferation rate was markedly reduced but increased differentiation into osteoblasts and adipocytes was observed [98]. These results indicate that elevation of intracellular Cu levels in healthy and cancer cells induced maturation of undifferentiated progenitor cells towards a specific differentiation pathway. 

In contrast to healthy unmatured cells, whose proliferation was inhibited by Cu salts, undifferentiated cancer cells demonstrated increased cell proliferation when treated with non-toxic concentration of CuCl_2_ (up to 100 μM); however, their differentiation was also stimulated [99]. In agreement with these observations, cancer cells that were treated with the differentiating agents, such as retinoic acid, demonstrated significantly higher intracellular Cu levels [100], and differentiation was augmented by Cu addition [101] and decreased by pre-treatment with Cu chelators [102]. 

Although non-toxic Cu load was shown to stimulate proliferation of cancer cells, the overload of intracellular Cu pool has eventually led to programmed cancer cell death in a broad panel of cancer cell lines [103]. Interestingly, transcriptional profiling revealed that cell death in these cell lines occurred as a result of endoplasmic reticulum (ER) stress and Unfolded Protein Response (UPR), but not ROS-induced oxidative stress, as would be intuitively expected. Moreover, GSH, which is a key survival antioxidant, was shown to protect cells from Cu-induced effects mainly via direct binding to Cu ions, thereby preventing protein misfolding or aggregation [103].

### 3.4. Metabolism

Hanahan et al. demonstrated that cancer cells treated with a Cu chelator were characterized by the reduced proliferation rate, which was linked to the reduced ATP production in mitochondria, as well as diminished activity of the electron transport chain (ETC) [57]. Interestingly, ATP production in cancer cells was partially restored via aerobic glycolysis and increased lactate production, indicating pro-survival compensatory attempts. These fundamental findings revealed the effects of Cu on metabolic phenotype of the tumors and demonstrated the dependence of Cu-rich cancer cells on oxidative phosphorylation (OXPHOS) and Cu-deficient cancer cells on glycolysis. 

Several omic studies revealed the effects of modulation of Cu levels on cellular proteomic and transcriptomic profiles and the role of Cu in the metabolic switch from oxidative phosphorylation to aerobic glycolysis [104,105,106]. It was shown that in Chinese hamster ovary cells (CHO) Cu deficiency induced mainly post-transcriptional changes, characterized by a compromised expression of COX, whose active site contains two Cu centers (3 atoms) and disrupted ETC, in particular mitochondrial complex IV [104,106]. The subsequent metabolomics studies in Cu-deficient cancer cells supported omic studies and confirmed the switch to aerobic glycolysis via disruption of ETC [107,108]. It was shown that Cu deficiency was associated with reduced efficiency of tricarboxylic acid cycle (TCA), excess lactate production, etc. [33]. The hypothesized sequence of events is presented in Figure 2 based on the obtained metabolic landscape [108]. 

The addition of non-toxic concentrations of Cu to healthy and cancer cells also revealed marked changes in cell metabolism. Incubation of CHO cells with 5 μM of CuSO_4_ revealed differential expression of genes involved in cell metabolism, in particular ATP synthesis, lactose dehydrogenase, and oxidative stress [105]. Similarly, exposure of human erythroleukemic cells K562 to 100 μM of CuCl_2_ significantly altered mitochondrial bioenergetics and boosted OXPHOS capacity [99] in agreement with the observations of Hanahan et al. [57]. In addition, following the increase in ATP demand, mitochondrial volume, size, and turnover have been adjusted, indicating tight metabolic control of mitochondrial energy by bioavailable Cu [105]. 

The Cu addiction in metabolically active tumors is supported by correlations between high Cu levels and metabolic demands of prostate, breast, colon, liver, and brain tumors [109,110,111,112,113,114]. Therefore, these cancer types may be particularly sensitive to Cu-modulating therapies. Recently, it was demonstrated that metabolic adaptions play an important role in highly resistant colorectal cancers with KRAS (Kirsten rat sarcoma 2 viral oncogene homologue) mutation [115]. Strikingly, these profound metabolic dependencies were also linked to the addiction of KRAS-mutated cancer to Cu metabolism [116,117]. Profiling of KRAS-expressing intestinal epithelioid cells IEC-6 with the combination of state-of-the-art cell surface proteomics and CRISPR/Cas-9 screens identified ATP7A as a lethal synthetic partner in KRAS-mutant colorectal cancers [116]. Subsequent target validation revealed the dependence of KRAS-mutant cells on bioavailable intracellular Cu pools regulated by micropinocytosis; therefore, Cu-targeting drug molecules might have clinical potential in the context of KRAS-mutated cancers. 

### 3.5. Resistance

The role of Cu in drug resistance is mostly discussed in relation to Cu transporters (Ctr1, Ctr2, OCT2), which play a significant role in resistance and detoxification of Pt-based chemotherapeutic agents [118]. However, Cu itself is also an important determinant of tumor resistance to chemotherapeutic treatment. When leghorn embryos were injected with CuSO_4_, Adriamycin, or their combination, Adriamycin-treated embryos demonstrated inappropriate development of the neural tube, while the addition of Cu salt has annihilated the effects of Adriamycin, thus resulting in normal embryo development similar to controls [119]. These results indicate that rapidly proliferating tissues developed drug resistance in the presence of high Cu concentrations. It was also reported that mice bearing cyclophosphamide-resistant Lewis lung carcinoma or doxorubicin-resistant Ehrlich ascites carcinoma were characterized by elevated serum Cu levels compared with respective drug-sensitive parental tumors [120]. Importantly, similar correlations have been observed in a clinical setting. When serum Cu levels were determined in patients with advanced breast, lung, and colon cancers that were undergoing chemotherapy, clinical non-responders demonstrated significantly higher serum Cu concentrations than patients who positively responded to the treatment [120]. 

Since Cu is a redox-active metal [121], it is highly toxic to cells in excess concentrations. Therefore, the elevation of intracellular Cu concentrations is counteracted in cells by various survival mechanisms to rapidly prevent excessive Cu accumulation. One of the mechanisms involves the downregulation of Ctr1, which is the major Cu influx transporter [122,123]. It was shown that low micromolar Cu concentrations were sufficient to stimulate the process of endocytosis in HEK293 cells, whereas high Cu concentrations induced degradation of the transporter [122]. Ctr1 is a major mediator of cisplatin and carboplatin uptake; therefore, its endocytosis and degradation were linked to cancer cell resistance to these Pt chemotherapeutic agents, as well as poor prognosis [24,124]. Besides influencing the influx of Pt drugs, Cu levels were also reported to affect Pt efflux via ATP7A and ATP7B, which are major Cu efflux transporters [124]. In the study of Komatsu et al., human epidermoid KB carcinoma cells that were transfected with c-DNA of *ATP7B* and subsequently treated with increasing concentrations of CuCl_2_, demonstrated increased survival in the presence of cisplatin, suggesting increased efflux of this Pt drug [125]. 

Interestingly, elevated Cu concentrations were also shown to affect plasma proteins that are responsible for the delivery of Cu ions to plasma membrane, in particular human serum albumin (HSA) [126] and bovine serum albumin (BSA) [103]. HSA was treated with physiologically relevant Cu concentrations for 21 days, corresponding to average Cu concentrations in healthy people and cancer patients, and the structure of HSA was analysed by various methods [126]. It was shown that increased Cu concentrations caused structural alterations of HSA, indicating activation of UPR, which is another pro-survival cellular response [127]. These results were corroborated using BSA [103]. It was reported that HSA aggregation adversely affected its drug-binding ability, hence, Cu-induced structural changes of HSA might also contribute to drug resistance [128].

### 3.6. Mode of Cell Death

The excess of Cu ions in cancer cells results in the disturbance of controlled redox environment, as well as overload of Cu-handling capacity of proteins. This leads to the perturbation of their structures, accumulation of misfolded or unfolded proteins, and activation of UPR [127,129,130]. As a result of severe ER stress, cells undergo vacuolization, resulting in a unique mode of cell death termed paraptosis. Paraptotic cells lack the typical morphological features of apoptotic cells, such as chromatin condensation and nuclear fragmentation. Treatment of various cancer cell lines with CuCl_2_ at toxic concentrations (>200 μM) for 48 h resulted in cancer cell death, reflected by massive vacuolization [130]. The vacuolated cells were not characterized by caspase-3 cleavage, which is the most characteristic feature of apoptosis. It was suggested that Cu ions might inhibit caspase-3 protease by interfering with catalytic residues on the active site of the enzyme [130] similar to Zn(II) ions [131]. Various Cu-based drugs demonstrated similar morphological changes in dying cancer cells, indicating that cancer cells undergo Cu-induced paraptosis independent of the presence of organic ligand [129,130,132,133,134]. In contrast to the effects of Cu overload, Cu depletion was associated with significant decrease of tumor burden as a result of caspase-3 activation and apoptotic cell death [38]. In agreement, the addition of CuSO_4_ to a panel of cancer cell lines in combination with various Cu chelators in a ratio 1:1 or 1:2 caused rapid apoptosis within 20 min [135]. 

Upon occurrence of severely stressful conditions, cells might concurrently undergo different types of cell death. Simultaneously with paraptosis, a fraction of Cu-treated cancer cells were shown to undergo apoptotic cell death [130]. It should be noted that rapid exposure to highly toxic concentration of Cu(II) ions might also cause necrotic cell death both in vitro and in vivo [136]. In healthy cells, Cu salts were shown to indirectly induce apoptotic cell death via inhibition of X-linked inhibitor of apoptosis protein (XIAP) [137]. HEK293 cells were treated with non-toxic concentrations of CuSO_4_ or CuCl_2_ for 48 h, which resulted in the destabilization of XIAP by interaction with its cysteine residues. The destabilization could be reversed by the addition of Cu-chelator [137,138]. Similar effects were observed in vivo in liver tissues from dogs with Cu toxicosis [137]. In cancer cells, CuCl_2_ also induced mobilization and destabilization of XIAP; however, this resulted in the enhancement of its activity and inhibition of apoptosis [38]. We speculate that the mode of cell death induced by Cu-based drugs might depend on the concentration of unbound Cu ions. It was reported that even trace amounts of free Cu ions completely abrogated caspase-3 activity, while protein-bound Cu in the absence of free Cu ions stimulated the sensitization of cells to caspase-3-dependent apoptotic cell death [137]. We have previously demonstrated that several Cu(II) complexes induced XIAP inhibition and apoptosis activation [139]. 

### 3.7. Immune System

The role of Cu in regulating cancer immune function has rarely been explored. In 1981, long before “evasion of immune system” was established as a cancer hallmark, the effects of Cu on the inflammatory responses in mice were reported [140]. Mice were fed either a purified Cu-deficient diet or normal diet with drinking water supplemented with CuSO_4_ and the differences in intracellular Cu uptake were reflected by ceruloplasmin and liver Cu levels [140]. The immune response was stimulated by exposure to foreign antigens, such as sheep erythrocytes. As a result, it was shown that Cu-deficient mice generated significantly less antibody-producing cells, indicating severely impaired immune system. Similarly, Cu-depleted rats showed reduced antibody titers [141]. In another study mice were given either purified Cu-deficient diet or Cu-rich diet supplemented with CuCl_2_ for 7–9 weeks after weaning [142]. It was demonstrated that after histamine injection Cu-deficient mice developed an inflammatory response reflected by paw oedema. Similarly, Cu-deficient mice demonstrated enhanced hypersensitivity to oxozalone and sheep red blood cells in comparison to mice supplied with CuCl_2_ [142]. The hypersensitivity test aims to demonstrate the response to histamine release; therefore, the results indicated that endogenous Cu was involved in the mediation of inflammatory responses [142]. In agreement, Cu-deficient mice, sheep, and cattle were also more susceptible to bacterial and microbial infections and were characterized by decreased leucocyte, erythrocyte and lymphocyte function, attributed to the compromised function of superoxide dismutase and COX [143,144,145,146,147]. Similarly, phagocytic activity of macrophages, which are essential for the activation of T and B cells, was negatively affected by Cu depletion [148]. Peritoneal macrophages from Cu-deficient rats opsonized markedly less yeast cells than from rats supplemented with CuCl_2_ [148]. This compromised phagocytic activity could be reversed by Cu supplementation of Cu-depleted calves [143]. 

Recently, the role of Cu in eliciting antitumor immune responses has been revealed. Analysis of Cancer Genome Atlas database and tissue microarrays demonstrated a statistically significant correlation between Ctr1 and programmed death ligand 1 (PD-L1), responsible for control of adaptive cancer immunity [149]. To support this observation in vitro, SH-SY5Y neuroblastoma and U87-MG glioblastoma cells were cultured in a media supplemented with 20 μM-1mM CuCl_2_ and PD-L1 expression was correlated with intracellular Cu content. It was demonstrated that Cu-treated cells induced dose-dependent increase of PD-L1 expression (at mRNA and protein level). In addition, RNA sequencing revealed activation of major signaling cascades involved in immune response, namely cytokine response or cytokine production, JAK-STAT signaling, and NFkβ-mediated TNFα signaling. Subsequently, inhibition of JAK-STAT signaling pathway was validated in cells treated with Cu-lowering drugs, such as Dextran-Catechin (DC), inhibiting Cu uptake, and Cu-chelating agent tetraethylenepentamine pentahydrochloride (TEPA). Moreover, this treatment induced inhibition of epidermal growth factor receptor (EGFR), leading to the destabilization and post-translational degradation of PD-L1. Similar results were obtained in vivo in the immunocompetent transgenic neuroblastoma mouse model *Th-MYCN*, which closely recapitulates the main features of poor outcome human neuroblastoma, as well as immunocompromised mouse model. Only immunocompetent mice demonstrated improved survivability and tumor reduction following oral administration of DC and TEPA, indicating the role of immune system in the drug-induced anticancer effects. The immune-competent mice were characterized by significant down-regulation of PD-L1 expression coinciding with elevated tumor infiltration by CD8^+^ tumor-infiltrating lymphocytes and natural killer (NK) cells. The levels of interferon-gamma (IFNγ), which governs PD-L1 expression, were also elevated in response to TEPA, suggesting that Cu-chelating drugs might be able to counteract the IFNγ-mediated anticancer immune reaction. Importantly, in contrast to tumor tissues, TEPA did not show any effects on PD-L1 and Ctr1 expression in normal livers, demonstrating certain selectivity towards cancer cells and their potential as immune checkpoint inhibitors [149]. In another study, the effects of Cu supplementation on PD-L1 expression and tumor reduction were monitored in immunocompetent and immunodeficient mice with hepatocellular carcinoma [150]. The tumor growth in mice with functional immune system was not affected by additional Cu supply, while immunodeficient mice demonstrated significant reduction of tumor growth. These effects were linked to Cu-induced stabilization of PD-L1 and T-cell suppression. This observation is of clinical importance since the effects of Cu-based anticancer drugs might be annihilated by the tumor-immune system. 

## 4. Modulating Intratumoral Cu Levels with Metal-Binding Ligands

Intracellular Cu balance is tightly regulated and both elevation and reduction of bioavailable Cu might cause interference with cancer development and progression as discussed above. This concept is utilized in therapeutic approaches aiming to deplete cancer cells of Cu supply or stimulate fatal Cu overload, yet these effects are commonly associated with toxicity. 

The control of intracellular Cu levels can be ensured by the use of Cu-binding organic ligands, which are classified based on the functional outcome of the resulting Cu complex [151]. 

### 4.1. Cu Chelators: D-Penicillamine, Trientine, and Tetrathiomolybdate

The most common mechanism of Cu chelators involves intracellular chelation of Cu ions with subsequent excretion of the metal. Cu chelators typically form Cu complex after they cross cell membranes, and the increase of Cu concentration (e.g., in cell culture media) is expected to lower the efficacy of Cu chelators, which are in agreement with their proposed mechanism of action [152]. In addition, it was shown that administration of Cu chelators during meals effectively prevented Cu intestinal absorption [153]. The use of Cu chelating agents for lowering body Cu levels has been extensively investigated in relation to medical conditions caused by Cu overload, such as Wilson’s disease [154]. The first clinically approved Cu chelator was D-penicillamine (D-pen), an analogue of cysteine aminoacid (Figure 3A). The mechanism of action of D-pen is based on the chelation of Cu(II) ions (Figure 3B) with subsequent formation of stable Cu(I)/oxidized D-pen complex and its subsequent excretion through urination [155]. Various clinical trials utilizing D-pen for the treatment of Wilson’s disease revealed severe toxicity induced by negative Cu balance (depletion of several milligrams of Cu per day), including bone marrow suppression, various dermatological side effects, autoimmune reactions, and neurotoxicity [156,157]. Despite low cytotoxicity in vitro [135], D-pen demonstrated significant reduction of primary tumor burden, as well as metastases, in various animal models, which was related to its antiangiogenic effects [158,159,160,161,162]. Based on the excellent efficacy of D-pen in brain tumor models, clinical trials were initiated where D-pen was administered to glioblastoma multiforme patients in combination with a low-Cu diet (e.g., NCT00003751, Table A1) [163]. The treatment was well-tolerated and effective decrease of serum Cu levels was observed; however, no significant improvement in survival was reported [163]. The discrepancies between poor effects of D-pen in a brain animal model and brain cancer patients might be related to several factors. In animals, higher dose of D-pen was given. Moreover, combination of D-pen and low-Cu diet was given before implantation of brain cells, suggesting that Cu depletion might be more effective in prevention of angiogenic switch rather than suppression of highly vascularized tumors [161]. Therefore, the effects of D-pen might be more pronounced in patients with low-grade gliomas or anaplastic astrocytomas. 

Due to severe side-effects induced by D-pen treatment in some Wilson’s patients, another Cu(II)-selective chelator, namely triethylenetetramine (trientine, TETA), characterized by an improved safety profile, was introduced into clinical practice (Figure 3A) [164]. Its anticancer activity was linked to the inhibition of angiogenesis, inhibition of Cu/Zn superoxide dismutase, and telomerase inhibition [164]. In clinical trials where trientine was used in combination with carboplatin or PEGylated liposomal doxorubicin for the treatment of relapsed ovarian, tubal, or peritoneal cancers, no dose-limiting toxicity has been observed; however, no correlations between clinical response and Cu or ceruloplasmin levels were reported [165]. Based on the promising in vitro and in vivo effects of trientine in cancers carrying BRAF^V600E^ mutation, another phase I clinical trial was initiated, where trientine was tested in combination with BRAF^V600E^ inhibitor Vemurafenib; however, this clinical study has been subsequently withdrawn (NCT02068079, Table A1). One of the main disadvantages of trientine, which hindered its clinical success, might be attributed to its poor adsorption and extensive metabolic transformation into *N_1_*-acetyltriethylenetetramine (MAT) and *N_1_,N_10_*-diacetyltriethylenetetramine (DAT) [166]. 

Another extensively studied Cu chelator is TM (Figure 3A) [156]. The discovery of its Cu-binding properties dates back to 1943, when ruminant animals, in particular milking cows, in the farms of New Zealand and Australia developed a medical condition which was later associated with Cu deficiency [167]. It was found that animals were fed on grass from Mo-rich soil. Mo reacted with disulfide from cellulose in grass in the rumen, forming thiomolybdates [168,169]. This hypothesis was supported by observation that non-ruminant animals did not develop any Cu-deficiency symptoms [167]. Subsequently, it was revealed that TM-induced Cu-deficiency in rats could be restored by Cu supplementation [170]. Moreover, intravenous administrations of TM into sheep with Cu poisoning reversed the deleterious effects [171]; therefore, TM was investigated for the diseases associated with Cu overload, such as Wilson’s disease [156]. In the bloodstream TM readily binds to serum albumin or other Cu-transporting macromolecules, thereby forming tripartial protein/TM/Cu complex (Figure 3B) rendering Cu unavailable for cellular uptake [172]. Subsequently, TM demonstrated anticancer activity in vitro and in vivo, which was related to its Cu-chelating properties [156]. Administration of TM into RIP1-Tag2 mice, which spontaneously developed pancreatic cancer (Figure 1), delayed the onset of angiogenic switch [57], indicating antiangiogenic properties of TM [57,173]. In phase I clinical trial, TM-induced restoration of normal Cu level in cancer patients without additional toxic effects [174]. In phase II clinical trial patients with advanced kidney cancer demonstrated a high tolerability of TM treatment; however, only a small proportion of patients achieved a steady disease state (Table A1) [175]. In another phase II clinical trial Cu depletion induced by TM did not affect the progression of cancer in hormone-refractory prostate cancer patients [176]. Phase II study in patients with stage II–IV breast cancer demonstrated that patients with triple-negative breast cancer (TNBC) were more susceptible to TM treatment than patients with other breast cancer subtypes [177]. This is in agreement with higher Cu demand of metabolically active tumours as discussed above [57]. It was suggested that efficacy of TM in TNBC patients might be linked with targeting tumor microenvironment, in particular endothelial progenitor cells, critical for angiogenic switch [177]. Overall, various clinical trials demonstrated that TM treatment is relatively well tolerated, but its efficacy as sole therapy might be limited, especially in large or metastatic tumors [154,156] and it might be more effective in a combination with cytotoxic chemotherapeutic schemes or other antiangiogenic drugs. For more detailed information on preclinical studies and early clinical trials with anticancer Cu chelators please refer to the following reviews [178,179].

Another example of a promising anticancer drug candidate with metal-chelating properties is the natural product trodusquemine, also known as MSI-1436 (Figure 4A). It was isolated from the dogfish shark by Magainin Pharmaceuticals Inc. [180] and demonstrated a wide range of biological activities [181,182]. In particular, it was shown to selectively and reversibly inhibit protein tyrosin phosphatase 1B (PTP1B) [182], which is a validated target for obesity and diabetes [183], and its high expression is associated with poor prognosis in different types of cancer [184,185]. As a consequence of PTP1B inhibition, MSI-1436 significantly inhibited the growth of primary breast tumors, as well as significantly decreased the development of lung metastases [182]. The safety and tolerability of MSI-1436 by metastatic breast cancer patients was tested in a phase I clinical trial (NCT02524951, Table A1); however, the study was terminated due to funding issues. To improve oral bioavailability of MSI-1436, more potent analogue termed DPM-1001 was developed [186]. Similar to MSI-1436, DPM-1001 demonstrated inhibition of PTP1B; however, in the case of DPM-1001 the inhibition was Cu-dependent. In presence of Cu(II) salts DPM-1001 formed PTP1B-Cu(II) complex, which was a more potent inhibitor of PTP1B than uncoordinated ligand (Figure 4B) [186]. There are several examples of metal-based phosphatase inhibitors [187]; however, in contrast to PTP1B-Cu(II), their mechanism of phosphatase inhibition is not clear. 

### 4.2. Cu Ionophores

In contrast to Cu chelators, Cu ionophores chelate metal ions in the extracellular space then transport them via biological membranes and release them in the intracellular space, thereby increasing intracellular Cu concentration. Subsequently, Cu ionophores recirculate into the extracellular space or reside in the cell membrane. At this point, the increased Cu concentration would lead to the enhanced activity of the metal-binding ligand [152]. The activity of Cu ionophores is Cu-dependent and metal-free ligands in Cu-depleted conditions are either inactive or significantly less active than in the presence of Cu ions [188]. The most extensively studied organic Cu-binding ligands that could act as Cu ionophores include disulfiram, clioquinol, elesclomol, pyrithione, and thiosemicarbazones. For more detailed information please refer to the review of Franz et al. [151].

#### 4.2.1. Disulfiram and Dithiocarbamates

Disulfiram (DSF) is a clinically approved drug for treatment of alcoholism which can be orally administered. It is believed to act as an irreversible inhibitor of acetaldehyde dehydrogenase enzyme, which in the presence of alcohol triggers the release of acetaldehyde, leading to physical discomfort and vomiting [189]. The anticancer potential of this drug was discovered in 1977, in which a severely alcoholic patient with a metastasized breast tumour went into spontaneous remission while undergoing disulfiram treatment [190]. Numerous clinical trials have since been conducted [191], which demonstrated the significant therapeutic potential of DSF against various types of cancer. The mechanism of action of DSF is believed to be based on its ability to convert into diethyl dithiocarbamate (DDC) in stomach with subsequent binding to extracellular Cu ions (Figure 5A). This is supported by the increase of intracellular Cu content followed by incubation with DSF [188]. It was reported that the activity of DSF in vitro was potentiated by the addition of Cu salts and inhibited by the addition of cell non-permeable Cu chelator, in agreement with its ionophoric properties [188,192,193]. Therefore, in clinical trials, DSF is typically given to cancer patients in combination with a dietary Cu pill (e.g., NCT03323346, NCT04265274, NCT04521335, Table A1). 

The reaction of DSF with Cu ions has been extensively studied by a wide range of analytical methods. It was suggested that when cancer cells were treated with DSF in a culture media supplemented with Cu salts, DSF possibly underwent rapid Cu-induced oxidation with the formation of bis(dialkyliminium)-tetrathiolane dication as an intermediate (bitt-4^2+^), which was followed by the reduction of another DSF molecule into the Cu(DDC)_2_ complex (Figure 5B) [188,194]. As a result, cells would rapidly be exposed to deadly levels of ROS, leading to oxidative stress, ER stress, inhibition of proteasome and transcription factors, and cancer cell death [195,196]. It is unclear whether such in vitro mechanism might be translated to in vivo. In fact, even when well tolerated, DSF/Cu did not demonstrate significant clinical benefits in cancer patients with solid tumors when it was administered as a monotherapy (Table A1) [196]. However, DSF significantly potentiated the anticancer activity of other chemotherapeutic drugs, such as cisplatin, oxaliplatin, imatinib, etc. [196,197]. Therefore, the majority of currently undergoing DSF-related clinical trials employ DSF as a component of combination therapy. In particular, the use of DSF might be beneficial for treatment of brain malignancies (Table A1) [193,198,199], partially due to its ability to alter Cu biodistribution and metabolism, leading to the increased Cu concentrations in brain [200]. 

Structure–activity relationships revealed that thiuram fragment was essential for ROS generation and overall in vitro anticancer potency of DSF [201]. Methylation of a thiol group of DDC resulted in the complete inhibition of Cu binding and anticancer activity. These observations are important, since *S*-methyl DDC is the main liver metabolite of DSF [201]. Subsequently, various dithiocarbamates demonstrated Cu-dependent anticancer activity, including but not limited to dipyridylhydrazone dithiocarbamate (DpdtC) [202], pyrrolidine dithiocarbamate (PDTC) [203,204] and its analogue 1 [205] as well as many other derivatives (Figure 6) [206].

#### 4.2.2. Clioquinol and Hydroxyquinolines

Clioquinol (CQ) is another repurposed drug which demonstrated anticancer efficacy linked to its ionophoric properties. CQ was widely used as a component of an antimicrobial topical cream for the treatment of skin infections; however, it was eventually banned in many countries due to severe cases of optic nerve damage [207]. When CQ was administered orally it demonstrated promising therapeutic potential for the treatment of Alzheimer’s and Parkinson’s diseases and was progressed into clinical trials [208,209]. It was shown that Alzheimer’s patients undergoing CQ treatment were characterized by markedly reduced formation of amyloid plaques. The activity of CQ was postulated to be linked to its metal-chelating properties. CQ is a derivative of 8-hydroxyquinoline and it can chelate Cu, Zn, and Fe in a bidentate fashion via the oxygen and nitrogen donor atoms (Figure 7). Subsequently, it was hypothesized that CQ might be effective as an anticancer drug and display cytotoxicity as a result of metal chelation in cancer cells [114]. Interestingly, CQ-induced cytotoxicity was enhanced by supplementation of Cu, Zn, and Fe, arguing towards ionophore properties [210]. In agreement, CQ-treated mice demonstrated significant increase of Cu content in soluble brain and liver fractions in comparison to untreated control group [211]. However, when mice with human prostate tumor xenografts were treated with CQ, the tumors, analysed by synchrotron X-ray imaging, revealed that CQ led to a strong decrease of intratumoral Cu levels [114]. Simultaneously, no changes in Cu levels in healthy tissues has been observed, indicating high selectivity of CQ towards cancer cells with higher intracellular Cu content. It was suggested that such drastic decrease of intratumoral Cu levels might be related to the decreased demand for Cu in drug-treated tumors or decreased Cu accumulation due to reduced angiogenesis [114]. Importantly, it was revealed that CQ significantly increased Cu(II) tissue content, even though Cu(I) is a predominant form of Cu in cancer and healthy tissues [114]. This observation might indicate that Cu(II)-CQ complex might be the intracellular metabolite responsible for cancer cell death. The formation of Cu(II)-CQ complex might occur intracellularly, if CQ can replace Cu from its chaperones due to higher binding affinity or bind uncoordinated intracellular Cu(II) ions. The formation of Cu(II)-CQ might also occur extracellularly with subsequent transportation into the cells [114].

Similar to DSF, the activity of CQ was potentiated by Cu and linked to proteasome inhibition and oxidative stress [204,210,212]. The mechanism of action was similar to CQ and 8-HQ [213]. Global proteomic profiling of RAW264.7 macrophages treated with 8-HQ in the presence or absence of CuSO_4_ revealed destabilization of the proteasome and redox changes only in the presence of Cu salt [214]. In addition, 8-HQ affected actin cytoskeleton, which might be essential for inhibition of cell motility and metastasis [214]. The high-throughput screening of repositioned drugs identified CQ and DSF as candidates potentiating the cytotoxic effects of cisplatin in multiple cell lines [215]. However, the clinical potential of CQ is limited, especially based on the reports of CQ-induced neurotoxic responses in animals and patients [207,208].

Functionalization of the 8-hydroxyquinoline (8-HQ) scaffold of CQ can significantly enhance the anticancer properties. CQ analogues, such as compounds **2** and **3** (Figure 7) have exhibited a wide range of improvement (1.5–26-fold) in activity against various cancer cell lines, which was further potentiated by the addition of CuCl_2_ [213,216]. The screening of focused library of 8-HQ derivatives demonstrated the importance of 8-HQ backbone for selectively targeting multidrug-resistant (MDR) cancer cells [217]. Moreover, it was shown that the sensitization of MDR was dependent on the expression of P-glycoprotein (P-gp) [217,218].

#### 4.2.3. Elesclomol and Derivatives

In contrast to DSF, CQ, and pyrithione, elesclomol (STA-4783) has not been repurposed but was discovered by Synta Pharmaceuticals Corp. as a result of high-throughput screening of their unique compound library against human sarcoma cell lines, using elevation of 70 kDa heat shock protein (HSP70) as a biomarker of oxidative injury [219]. Initially, structurally related molecule *N*′^1^,*N*′^3^-diethanethioyl-*N*′^1^,*N*′^3^-diphenylmalonohydrazide 4 (Figure 8A) was identified, which demonstrated anticancer activity in a nanomolar concentration range [219]. However, this compound was air-sensitive and chemically and metabolically unstable, hence chemical modification was required to improve its therapeutic potential. Upon establishing structure–activity relationships, elesclomol has been identified, which was stable and demonstrated 10-times improvement of cytotoxicity in comparison with the parent compound. The anticancer activity of elesclomol was linked to the formation of elesclomol-Cu complex with Cu(II) ions [220,221] from cell culture media (in vitro) or bloodstream (in vivo) with subsequent transportation of Cu(II) ions to mitochondria, reduction to Cu(I), and release of Cu ions, thus leading to oxidative stress [222]. The role of elesclomol as Cu ionophore was supported by the loss of its anticancer activity in the presence of cell non-permeable Cu chelator [222]. In addition, it was shown that elesclomol was readily excreted from the cells after releasing Cu ions. After 3 h of cell incubation with elesclomol, cells were washed-out and the levels of Cu and elesclomol were measured by BCA assay and LC/MS/MS, respectively. While elesclomol levels were significantly reduced after 1 h post-wash-out, intracellular Cu levels remained high, indicating the excretion of unbound elesclomol [222].

Recently, genome-wide CRISPR/Cas9 screening unambiguously identified a single gene responsible for Cu-dependent anticancer activity of elesclomol and DSF [223]. Initially, the anticancer activity of elesclomol was screened against a panel of 724 cancer cell lines and cell line sensitivity was correlated with the genomic features of these cell lines. It was found that the top correlated gene expression feature was the gene *FDX1* located in mitochondria and involved in Fe–S cluster formation. Subsequently, leukemic K562 cells were treated with two elesclomol analogues, OTA-5781 and OTA-3998 (Figure 8A), and two genome-wide CRISPR/Cas-9 deletion screens were conducted [223]. This screen allows for the determination of the genes whose deletion leads to the loss of drug activity. Similar to elesclomol, the resistance to OTA-5781 and OTA-3998 was determined by the deletion of a single gene—*FDX1*. The target was validated by the evidence of direct binding of elesclomol and recombinant FDX1, which was obtained by nuclear magnetic resonance (NMR) spectroscopy. However, it was shown that cell death induced by elesclomol was not caused by the interaction between FDX1 and elesclomol but rather between FDX1 and elesclomol-Cu(II) complex (Figure 8B). It was suggested that binding of FDX1 and elesclomol-Cu(II) complex resulted in the full oxidation of FDX1, thereby inhibiting Fe–S cluster biosynthesis and requiring the reduced form of FDX1. Importantly, neither elesclomol nor Cu(II) were capable of oxidizing FDX1, thereby confirming the role of Cu(II) complex formation in the anticancer activity of elesclomol and its analogues [223]. In agreement, Cu chelation, as well as the replacement of Cu-binding moiety in the structure of elesclomol, resulted in the loss of its cytotoxicity [223].

The antitumor efficacy of elesclomol was tested in a human melanoma and lymphoma xenograft models and no significant reduction of tumor burden has been observed [219]. However, elesclomol significantly potentiated the effects of paclitaxel both in vitro and in vivo [219]; therefore, it entered clinical trials in combination with paclitaxel for treatment of advanced metastatic melanoma (Table A1) [224,225]. The clinical trial was prematurely terminated when it was shown that the group of patients treated with combination of elesclomol and paclitaxel demonstrated a higher rate of total deaths than patients treated only with paclitaxel [225]. Subsequently, upon further analysis of the data, FDA approved the resumption of clinical trials, since clinical deaths were observed only in a small subset of patients with elevated levels of lactose dehydrogenase (LDH) and were not related to elesclomol-induced toxicity. In combination with paclitaxel, elesclomol demonstrated 1.8-fold improvement of progression-free survival in patients with normal LDH levels in comparison to paclitaxel alone. However, patients with high LDH levels showed no benefit.

Why was the activity of elesclomol governed by LDH levels? Elesclomol was shown to deliver Cu(II) ions selectively to mitochondria and interfere with mitochondrial function and metabolic processes that requires the presence of oxygen [222]. In contrast, the highest Cu accumulation in cells treated with DSF was observed in cytosol [222]. Under hypoxic conditions cells undergo metabolic reprogramming from OXPHOS to glycolysis, which mostly occurs outside mitochondria. Hypoxic conditions are characterized by high LDH levels, which explains the loss of elesclomol activity. Hence, in subsequent clinical trials, patients with elevated LDH levels were not eligible for elesclomol treatment (Table A1) [225,226]. The phase II clinical study exploited the combination of elesclomol and paclitaxel for treatment of recurrent ovarian and primary peritoneal cancers [226]. While this drug combination was well-tolerated, no significant clinical benefit was observed, hence, the clinical potential of elesclomol is not clear. However, recently it was suggested that cancer cells with resistance to proteasome inhibition were selectively sensitive to elesclomol and DSF [223]. It was shown that elesclomol significantly potentiated the activity of proteasome inhibitor bortezomib in multiple myeloma orthotopic mouse models; therefore, elesclomol might be clinically efficient in combination with bortezomib or other proteasome inhibitors.

#### 4.2.4. Thiosemicarbazones (TSCs)

TSCs represent a class of chelating ligands of a wide range of elements from Groups 5–15 in the Periodic Table with various bonding patterns [227,228]. This class of compounds demonstrated versatile biological properties and has been extensively investigated for several decades. TSCs can be broadly classified into mono- and bis-TSCs and their anticancer activity is described in details in other reviews [9,229,230,231,232]. One of the most well-studied TSCs is 3-aminopyridine carboxaldehyde TSC (3-AP, Triapine) designed by Sartorelli et al. [233,234], which is a potent Fe chelator interfering with intracellular Fe metabolism (Figure 9) [235].

3-AP has been employed in many clinical trials in combination with chemotherapy and radiotherapy where positive responses have been observed (Table A1) [230]. However, 3-AP treatment was commonly associated with severe toxicity and even treatment-related death; therefore, more selective TSCs have been developed, including di-2-pyridylketone TSCs (DpT series), 2-benzoylpyridine TSCs (BpT series), and 2-acetylpyridine TSCs (ApT series) (Figure 10) [229]. Following extensive structure–activity relationships, di-2-pyridylketone 4,4-dimethyl-3-thiosemicarbazone (Dp44mT) demonstrated more than 100-times increase of cytotoxicity and improvement of selectivity and therefore was chosen as the lead compound from the DpT series (Figure 9).

Interestingly, despite structural similarities, 3-AP and Dp44mT demonstrated drastically different responses to the supplementation of cell culture media with Cu(II) salts [236]. The activity of 3-AP was abrogated by the stoichiometric addition of CuCl_2_; however, the activity of Dp44mT was markedly enhanced. The loss of 3-AP activity was explained by the base-catalyzed desulfurization of 3-AP/Cu complex, leading to the formation of poorly soluble CuS and inactive nitrile (Figure 11) [230,237]. In contrast, terminal demethylation of amino group in Dp44mT prevented the loss of sulfur, as well as the absence of cytotoxicity. However, an alternative hypothesis ruled out the desulfurization of 3-AP based on the solution equilibrium studies, which demonstrated high stability of 3-AP/Cu complex [238,239].

Potentiation of Dp44mT activity in the presence of Cu ions indicated that Dp44mT might act as Cu ionophore and suggested Dp44mT/Cu complex as a biologically active form of Dp44mT in addition to the well-known Dp44mT/Fe species and 3-AP/Fe species [236]. Such differential metal ion dependency dictated distinctly different anticancer mechanisms of action of 3-AP and Dp44mT. The mechanism of action of 3-AP was mainly linked to the Fe chelation with the subsequent formation of redox-active 3-AP/Fe complex, followed by inhibition of ribonucleotide reductase (RNR), imbalance of dNTP pools, arrest of DNA synthesis and deadly ROS insult. On the other hand, the rapid cell killing effect of Dp44mT was associated with the quick formation of redox-active Dp44mT/Cu complex, which was followed by another mechanism associated with Fe depletion [236,240]. As a proof of concept, Dp44mT was administered to cancer cells in combination with increasing concentrations of Cu(II), and it was shown that cytotoxicity was proportional to intracellular Cu accumulation [241,242]. Interestingly, the differences in metal binding between 3-AP and Dp44mT predetermined differential treatment-associated adverse effects. The use of 3-AP caused severe hematologic malignancies associated with Fe chelation, such as anemia and methemoglobinemia [243,244]. In contrast, Dp44mT caused cardiac fibrosis [245], possibly related to ROS induction by Dp44mT/Cu complex. To avoid cardiac toxicity associated with Dp44mT, the structure was further optimized, resulting in the discovery of orally tolerated and metabolically stable di-2-pyridylketone 4-cyclohexyl-4-methyl-3-thiosemicarbazone (DpC) (Figure 9) [246,247]. DpC demonstrated high efficacy, tolerability, and favourable pharmacological properties, and entered multi-center clinical trials for treatment of solid tumors (NCT02688101, Table A1). Similar to Dp44mT, the activity of DpC was potentiated by Cu(II) and reduced by addition of cell non-permeable Cu chelator (Figure 9) [240,248,249]. It was shown that both Dp44mT and DpC accumulated in lysosomes and formed Cu complex with intra-lysosomal Cu, resulting in ROS production and lysosomal permeabilization [249,250]. The replacement of 2-pyridylketone fragment in DpT series with 2-benzoylpyridine in BpT series or 2-acetylpyridine in ApT series (Figure 10) revealed changes in redox potentials of Cu-TSC complexes [240]. However, all Cu(II) complexes were more active than the respective ligands, in agreement with the proposed mechanism of action [240,251]. Since lysosomes play an important role in autophagy, which is a major pro-survival cellular response, Dp44mT/Cu-induced lysosomal damage was also linked with impaired autophagic process and cell death [252]. The sequestration of Dp44mT to lysosomes was achieved by utilizing lysosomal drug transporter P-gp [253]. In agreement, Dp44mT demonstrated increased cytotoxicity in P-gp expressing MDR cell lines and tumors both in vitro and in vivo [253]. Although the role of lysosomal P-gp in drug resistance is not universally accepted [254], Dp44mT might be effective for the treatment of MDR tumors.

Another interesting example of a thiosemicarbazone with ionophoric properties is NSC319726 (ZMC-1, Figure 9), which belongs to the ApT thiosemicarbazone series. The synthesis and antimalarial properties of NSC319726 were first reported in 1979 [255]. Subsequently, this series of compounds demonstrated antitrypanosomal [256] antimycobacterial [257], antiviral [258], and antileukemic [259] properties. Almost 30 years later, NSC319726 was rediscovered upon bioinformatic analysis of the National Cancer Institute (NCI60) database. NSC319726, but not 3-AP, was shown to selectively target cancer cell lines with p53-mutated status (R175H mutation) [260]. Drug-induced reactivation of p53 was linked to restoration of Zn levels in mutant p53 in vitro and in vivo, leading to its correct folding and DNA binding [260,261]. Therefore, NSC319726 acted as a Zn metallochaperone (ZMC), whereas 3-AP was devoid of ZMC function [262].

NSC319726 was also shown to act as a Cu ionophore [263]; in fact, the binding affinity of this compound towards Cu(II) was several orders of magnitude higher than towards Zn(II) [264]. However, on the contrary to DSF, CQ, and elesclomol, whose activity was Cu-dependent, Cu binding was not necessarily beneficial for the anticancer activity of NSC319726. Interestingly, the restoration of p53 folding was compromised by ROS species induced by NSC319726/Cu complex, since oxidative damage activated post-translational modifications on correctly refolded p53, leading to cancer cell death and off-target toxicity [264]. As a consequence, combination of NSC319726 with ROS-inducing radiotherapy or cytotoxic chemotherapy did not demonstrate any synergism. However, when NSC319726 was pre-treated with GSH to quench unwanted ROS, increase of anticancer activity was observed [264]. Subsequently, extensive structure–activity relationship studies have produced analogues of NSC319726 with significantly diminished Cu binding, which demonstrated synergistic activity with cytotoxic chemotherapy even in the absence of antioxidants [265].

While ROS production was shown to interfere with the on-target mechanism of NSC319726 in p53^R175H^ cells, it might be beneficial in cancer cell lines with different type of p53 mutations or wild-type p53. When NSC319726 was screened against 13 primary glioblastoma patient-derived cell lines, it demonstrated excellent antiproliferative activity in nanomolar to picomolar concentration range, which was not significantly potentiated by Zn(II) supplementation but markedly potentiated by Cu(II) salts [263]. Transcriptomic profiling revealed the role of ROS induction as a mechanism of action of NSC319726/Cu(II) in patient-derived glioblastoma cell lines. These results indicate that Cu(II) ionophore properties are beneficial for the use of NSC319726 as a single agent and deleterious in combination therapy in p53^R175H^-mutant cancers.

Similar to NSC319726, another thiosemicarbazone, COTI-2, was shown to restore p53 activity [266,267,268]. It was discovered by Cotinga Pharmaceuticals Inc. as a result of high-throughput screening using computational CHEMSAS^®^ platform [269]. COTI-2 demonstrated high cytotoxicity in various cell lines, including highly resistant ones [270], and excellent antitumor efficacy with low toxicity in animal models [269]. In addition to targeting mutant p53, COTI-2 negatively modulated the PI3K/Akt/mTOR pathway independent of p53 status [267,268]. In contrast to NSC319726, COTI-2 did not affect intracellular Zn accumulation in HNSCC-mutant *TP*53 cells, indicating that its mechanism of action might be independent of Zn chelation [267]. The activity of COTI-2 was significantly potentiated by coordination to a Cu(II) center and decreased upon coordination to a Fe(II) center, suggesting that Cu plays an important role in the mechanism of action of COTI-2 [271]. Recently it was revealed that resistant profile of both COTI-2 and its Cu(II) complex was governed by GSH-dependent ABCC1 efflux pump [271]. It was shown that COTI-2/Cu complex formed stable and non-reducible adduct with GSH, which was recognized by ABCC1 and pumped out of the COTI-2-resistant cancer cells [271]. These findings are crucial for understanding potential resistance mechanisms in cancer patients. Currently, COTI-2 undergoes Phase 1, first-in-patient clinical trials in patients with recurrent hematological malignancies, head and neck squamous cell carcinoma, and other tumor types (NCT02433626, Table A1).

## 5. Modulating Intratumoral Cu Levels with Pre-Formed Cu Complexes

As was shown previously, the mechanism of action of chelators and ionophores is based on the formation of metal complexes. Chelators bind metal ions intracellularly, while ionophore bind metal ions in the bloodstream and release them once inside the cells. Therefore, prescribing patients with Cu supplements would potentiate anticancer activity of ionophores and diminish activity of chelators. However, even though ionophore/Cu complexes showed excellent in vitro results, the in vivo efficacy of ionophores in combination with Cu supplements did not demonstrate significant clinical benefit. One possible reason is due to the poor solubility of chelating ligands, which does not only hinder their intracellular accumulation, but also might inflict side effects associated with the use of co-solvents/surfactants. In addition, in clinics ionophores and Cu are not supplemented simultaneously and in the same solvent medium, which results in different pharmacokinetics, especially absorption in the bloodstream. Moreover, the uncoordinated ligand is available to bind alternative metal ions, which might possibly interfere with the on-target mechanism of action and induce off-target toxicity. A feasible approach is to administer pre-formed Cu complexes, thereby delivering the most favourable chelator/Cu ratio for optimal biological function.

### 5.1. Cu Complexes of Bis(TSCs)

This strategy was employed for the class of bis(thiosemicarbazones), whose wide range of biological properties was linked to their Cu, Fe, and Zn-ionophore properties. In particular, bis(thiosemicarbazone) (H_2_kts) demonstrated promising anticancer efficacy in vitro and in vivo when administered orally (Figure 12A). The activity of H_2_kts was diminished when rats bearing Walker 256 nitrogen-mustard-resistant carcinosarcoma were given a Cu-deficient diet and potentiated when H_2_kts was given together with Cu(II) supplements (approximate molar ratio of Cu(II):kts = 0.55) either in food or drinking water [272,273]. However, further increase of Cu(II) dosage (approximate molar ratio of Cu(II):kts = 1.1) has led to the off-target toxicity with no improvement of anticancer efficacy [272]. Moreover, Zn supplementation in the presence of Cu(II) reduced the efficacy of H_2_kts, while Zn-free environment (plastic animal cages, Zn-deficient diet) increased its efficacy [272]; therefore, in order to avoid interference with other metals, H_2_kts was chelated with Cu(II) to form Cu(kts) prior to administration into animals (Figure 12B) [274]. Cu(kts) demonstrated comparable reduction of tumor burden as the combination of H_2_kts with CuCl_2_ [274]; however, the pre-formed chelate was active irrespective of the presence of environmental or dietary Cu and less toxic (yet still toxic) due to the optimal ratio of Cu(II) to kts [274,275]. Subsequent in vitro experiments in sarcoma 180 ascites cells revealed that Cu(kts) was effectively accumulated in the intracellular space and subsequently eliminated from the cells after release of Cu ions as the result of complex reduction and thiol oxidation [276]. The mechanism of action of Cu(kts) was linked to the inhibition of DNA synthesis and mitochondrial respiration [277,278].

Extensive structure–activity relationships of Cu complexes of various bis(thiosemicarbazones) demonstrated the dependence of their activity and mechanism of action on their physicochemical properties. The more lipophilic complexes, such as Cu(ktsm2), were trapped inside the cellular membrane and were unable to release Cu ions as a result of reduction [279]. The complexes, which were more difficult to reduce intracellularly, were in general less active [280,281]. Comparison of structurally similar Cu bis(thiosemicarbazone) complexes, Cu(gtsm), and Cu(atsm), where the hydrogen atoms were replaced by two additional electron-donating methyl groups, demonstrated drastically different mechanism of action in relation to their distinctly different chemical behavior [282,283]. As expected, due to the presence of methyl groups, the reduction potential of Cu(atsm) was lower than that of Cu(gtsm), rendering Cu(atsm) more resistant to intracellular reduction and dissociation. This was confirmed using neuroblastoma SH-SY5Y cells transfected with a metal responsive element (MRE)-luciferase construct [283]. These cells were shown to respond to the increase of bioavailable Cu levels by the increase in luminescence [284]. When MRE-luciferase-transfected cells were treated with Cu(gtsm) and Cu(atsm), only Cu(gtsm) induced significant change in bioluminescence [283]. However, total intracellular Cu content was similar for both Cu complexes, indicating that increase of MRE response in Cu(gtsm)-treated cells was associated with the intracellular release of bioavailable Cu [283]. Subsequently, the anticancer effects of both complexes were tested in Transgenic Adenocarcinoma of Mouse Prostate (TRAMP) model and only Cu(gtsm) complex demonstrated the reduction of tumor burden and grade [282]. It was suggested that in vivo efficacy of Cu(gtsm) was related to the inhibition of proteasomal activity, which is typical for the increase of bioavailable Cu as described previously.

In contrast to cells with normal oxygen levels, under hypoxic conditions Cu(atsm) was shown to release bioavailable Cu [283]. Initially, it was hypothesized that hypoxia-induced Cu release was associated with selective Cu^II^/Cu^I^ reduction at lower pH, followed by release and deposition of Cu(I) [179], due to the impairment of ETC [283,285]. However, the retention of Cu(I) did not depend on intracellular oxygen concentration [179], hence, it was proposed that increased retention of Cu(I) under hypoxic conditions was linked to reduced reoxidation and efflux of the reoxidized species in comparison to the cells with normal levels of oxygen [231]. This selective hypoxia-induced retention of Cu(atsm) enabled its use as a hypoxia-specific positron emission tomography (PET) imaging agent. Currently, this drug candidate has shown successful outcomes from various clinical studies (Table A1) [231,286].

Preclinical development of Cu bis(thiosemicarbazone) complexes is hindered by their toxicity. The reported adverse side-effects in mice, rats, dogs, and monkeys included but were not limited to hematological abnormalities, renal toxicity, gastric ulcers, and loss of vision associated with cataract development [272,275]. Subsequently, several attempts have been made to modify the chemical structure of Cu bis(thiosemicarbazone) complexes aiming to reduce off-target toxicity [287,288]. An elegant approach termed “targeted ionophore-based metal supplementation (TIMS)” was proposed by Stahl and Chang et al. [288]. It is based on the conjugation of bis(thiosemicarbazone) with the ligand specifically recognized by a particular receptor, leading to the tissue-specific internalization of Cu complex with the following release of Cu ions for subsequent biological function. Cu(gstm) was conjugated with a liver-targeting *N*-acetylgalactosamine (GalNAc) group that is specifically recognized by the asialoglycoprotein receptor in liver (ASPGR) (Figure 12B). It was shown that Gal-Cu(gtsm) delivered significantly higher intracellular Cu content into cells and mice livers in comparison to untargeted Cu(gtsm). Strikingly, despite a marked increase of bioavailable Cu, treatment with Gal-Cu(gtsm) did not induce any Cu-related toxicity, which was reflected by normal liver H&E staining and serum liver enzyme levels [288].

### 5.2. Cu Complexes of 3-AP (Triapine) and Its Derivatives

Several attempts have been made to improve the cytotoxicity of 3-AP/Cu complex by derivatization of the 3-AP backbone. Terminal demethylation of amino group strongly enhanced the cytotoxicity of 3-AP [239] and synergism with CuCl_2_ [289]. It was shown that cytotoxicity and mechanism of cancer cell death was dependent on the stability of Cu(II)-TSC complexes and their reduction rates [290]. Complexes with micromolar activity, including 3-AP, were less stable and more prone to reduction by GSH, resulting in the liberation of metal-free ligands with their subsequent coordination to other metal ions, e.g., Zn from MTs or Fe from ribonucleotide reductase [139,290,291]. In contrast, complexes with nanomolar activity were characterized by the significantly higher stability and lower reduction rates and could probably reach intracellular target without dissociation. One possible biomolecular target of Cu(II)-TSC complexes is protein disulphide isomerase (PDI) located in the ER, which is associated with paraptotic cell death. It was shown that high stability of Cu(II)-TSC complexes was a prerequisite for efficient PDI inhibition and paraptosis [133,290].

Comparison of anticancer effects of metal-free TSCs with pre-formed Cu(II) complexes revealed crucial differences. Several Cu(II) complexes of 3-AP and its analogues did not demonstrate any improvement in cytotoxicity [188,292,293]. However, a large number of other pre-formed Cu(II)-TSC complexes were significantly more cytotoxic than their metal-free counterparts [139,290,294,295]. Moreover, the most cytotoxic Cu(II)-TSC complexes (nanomolar concentration range) demonstrated quick cancer killing activity (≈3 h), which was related to the rapid generation of the deadly ROS insult [290]. It was also suggested that the mechanism of action of less cytotoxic Cu(II)-TSC might be related to redox-independent pathways [232,290,295].

### 5.3. Drug Delivery Systems

Another approach to ensure simultaneous delivery of Cu-binding organic molecule and Cu salts is their encapsulation into various drug delivery systems, such as liposomes, micelles, and other nanomaterials. Different types of nanoformulations offer their own advantages, such as release rate, biocompatibility, ease of preparation, etc. It is believed that nanoparticles are able to selectively target malignant tissues due to Enhanced Permeability and Retention (EPR) effect. It is proposed that tumour blood vessels are characterized by the defective vasculature and nanoparticles within the certain size range might be extravasated from the leaky tumour blood vessels into the surrounding tumor tissues, while smaller molecules are cleared by the renal system. The EPR concept is debatable; however, several nanoformulations of anticancer drugs, such as Doxil [292], are currently in clinical use or undergoing clinical trials [293]. In order to ensure simultaneous delivery of Cu-binding ligands and Cu salts, various formulations have been developed. For example, a nanoformulation for simultaneous delivery of Cu and CQ has been recently developed, which demonstrated a significant increase of intracellular accumulation and anticancer efficacy [296]. Similarly, Cu-ionophore neocuproine was encapsulated with Cu(II) salt into thermosensitive PEGylated liposomes and the resulting nanoformulation demonstrated enhanced Cu accumulation and significant reduction of tumor growth in Balb/C mice [297]. Recently, selective mitochondria-targeting Cu-depleting polymeric nanoparticles (CDN) have been developed [298]. The design of CDN was based on the combination of a Cu-depleting moiety and a PEGylated semiconducting polymer fragment. These nanoparticles significantly inhibited mitochondrial respiration of TNBC cells, resulting in excellent in vitro and in vivo efficacy [298]. Moreover, intracellular Cu binding was followed by the decrease of fluorescent signal from the Cu-depleting moiety, thereby providing real-time feedback of the chelation [298]. This type of nanoformulation might hold a great promise for the treatment of aggressive cancers, such as TNBC; however, the selectivity of such nanoparticles towards cancer cells should be validated in vivo.

## 6. Conclusions

The disturbance of Cu homeostasis, either by Cu depletion or Cu overload, is a promising therapeutic strategy for cancer treatment. Various Cu-chelators, Cu-ionophores, and stable Cu complexes have been developed and some of them demonstrated great clinical potential, especially in combinations with cytotoxic modalities. However, due to the essential physiological function of Cu in the body, even slight changes in Cu concentrations are associated with significant toxicity. We have provided multiple examples where the clinical efficacy of Cu-binding or Cu-overloading drugs was compromised by either severe adverse effects or insufficient efficacy at well-tolerated dose levels. Moreover, the effects of the drug candidates were markedly dependent on the patient cohort, as well as the type of animal model. We believe that both Cu depletion and Cu overload might be viable therapeutic strategies; however, understanding of molecular vulnerabilities of specific cancer subtypes seems to be of fundamental importance and is absolutely necessary for clinical success of Cu-dependent drugs.

## Figures and Tables

**Figure 1 biomedicines-09-00852-f001:**
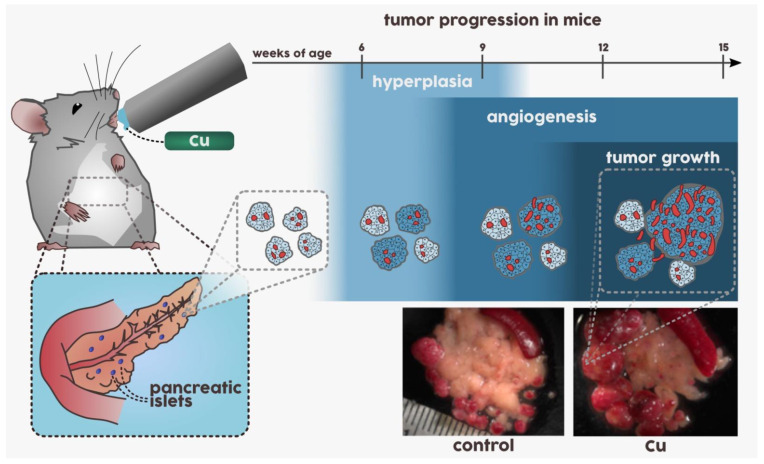
The RIP1-Tag2 mouse model of spontaneous pancreatic cancer development. Mice were given 20 μM of CuSO_4_ in drinking water from 4 weeks of age. Cu-treated mice demonstrated significantly higher volume of pancreatic tumors. Photographs of the tumors were taken from reference [57] with the permission from PNAS.

**Figure 2 biomedicines-09-00852-f002:**
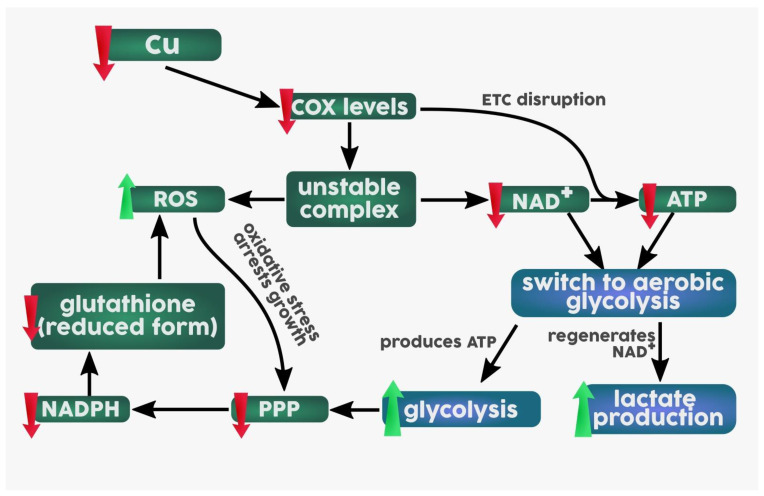
The effects of Cu-deficiency on the metabolic profile of CHO cells. Adapted from reference [108] (COX—cytochrome C oxydase, ETC—electron transport chain, ROS—reactive oxygen species, PPP—pentose phosphate pathway, NADPH—nicotinamide adenine dinucleotide phosphate, NAD^+^—nicotinamide adenine dinucleotide, ATP—adenosine triphosphoate).

**Figure 3 biomedicines-09-00852-f003:**
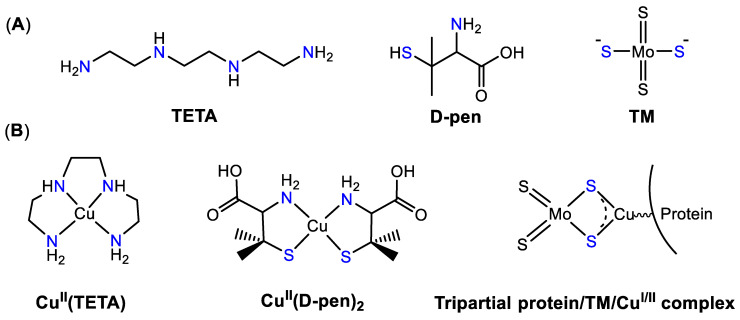
(**A**) Chemical structures of Trientine (TETA), D-penicillamine (D-pen), and tetrathiomolybdate (TM) and (**B**) their Cu complexes.

**Figure 4 biomedicines-09-00852-f004:**
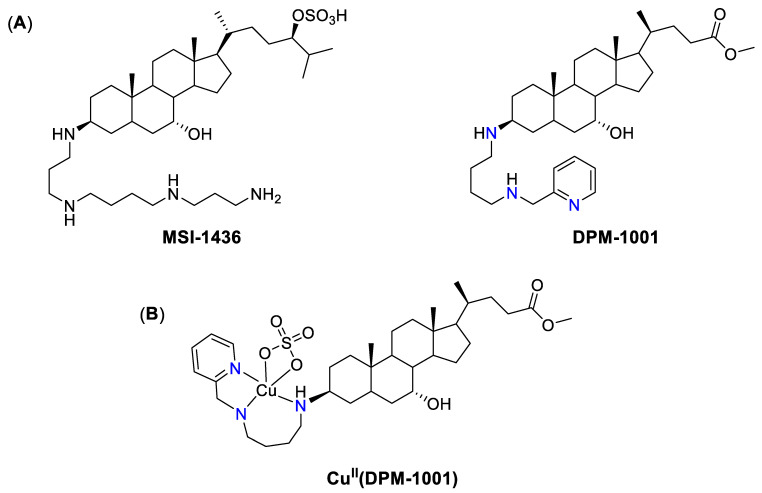
Chemical structures of (**A**) MSI-1436, DPM-1001, and (**B**) its Cu(II) complex.

**Figure 5 biomedicines-09-00852-f005:**
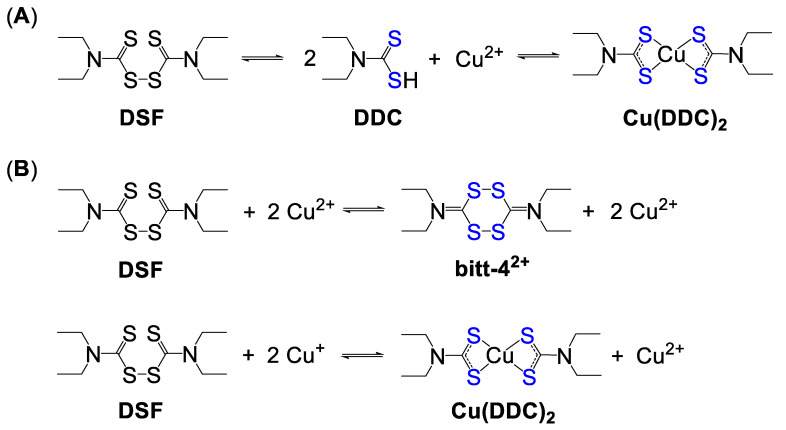
(**A**) Chemical transformation of disulfiram (DSF) in aqueous solutions and (**B**) mechanism with the formation of bitt-4^2+^ intermediate.

**Figure 6 biomedicines-09-00852-f006:**
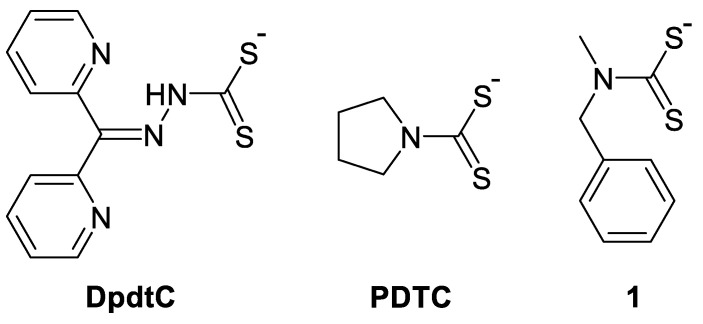
Chemical structures of dipyridylhydrazone dithiocarbamate (DpdtC), pyrrolidine dithiocarbamate (PDTC), and its analogue 1.

**Figure 7 biomedicines-09-00852-f007:**
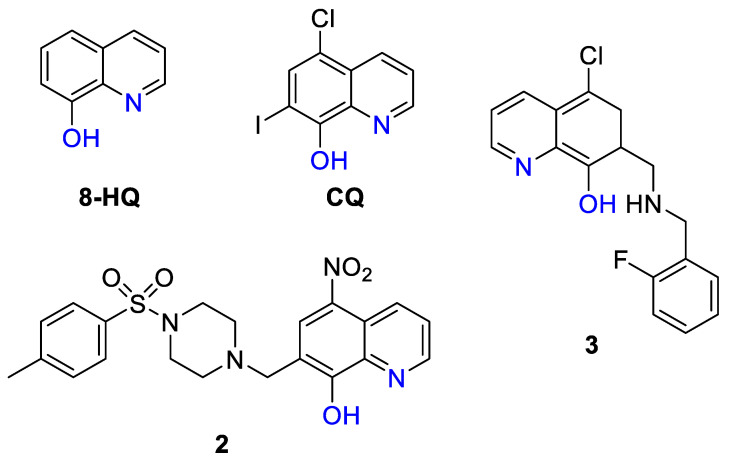
Chemical structures of 8-hydroxyquinoline (8-HQ), clioquinol (CQ), and its derivatives 2 and 3.

**Figure 8 biomedicines-09-00852-f008:**
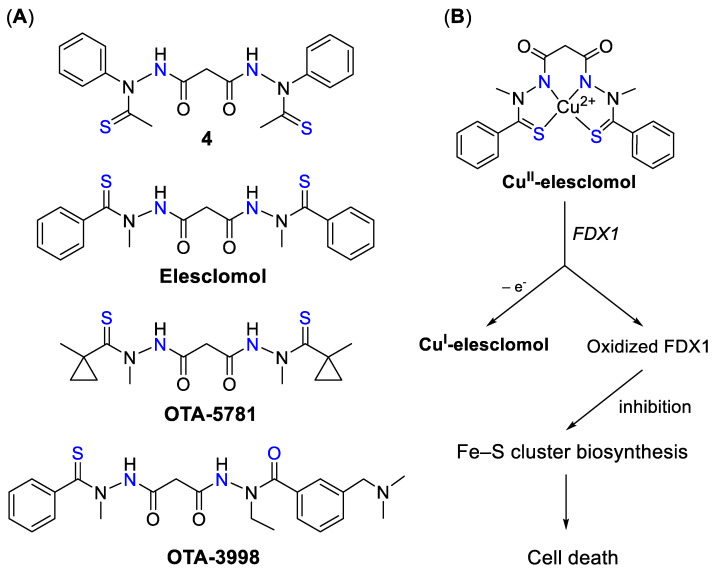
(**A**) Chemical structures of elesclomol, OTA-5781 and OTA-3998 and the parent compound of elesclomol *N*′^1^,*N*′^3^-diethanethioyl-*N*′^1^,*N*′^3^-diphenylmalonohydrazide (4); (**B**) proposed mechanism of FDX1-mediated, elesclomol/Cu-induced cell death induction.

**Figure 9 biomedicines-09-00852-f009:**
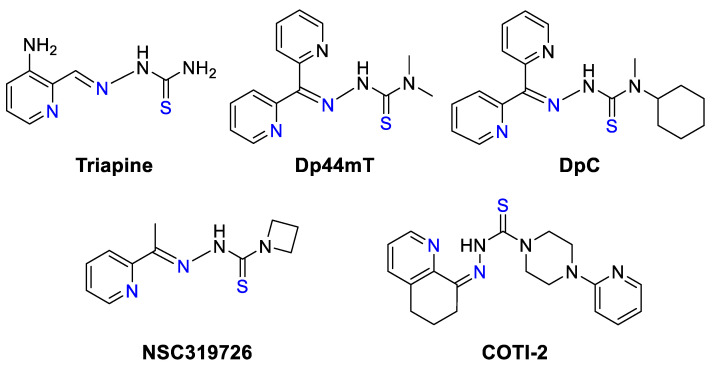
Chemical structures of Triapine (3-AP), di-2-pyridylketone 4,4-dimethyl-3-thiosemicarbazone (Dp44mT), di-2-pyridylketone 4-cyclohexyl-4-methyl-3-thiosemicarbazone (DpC), NSC319726 and COTI-2.

**Figure 10 biomedicines-09-00852-f010:**
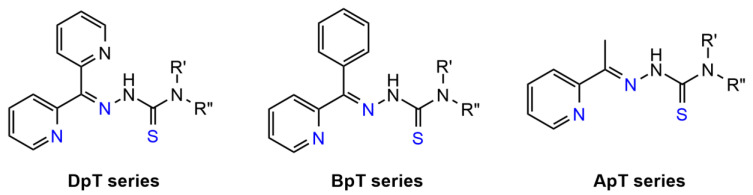
Chemical structures of di-2-pyridylketone TSCs (DpT series), 2-benzoylpyridine TSCs (BpT series), and 2-acetylpyridine TSCs (ApT series).

**Figure 11 biomedicines-09-00852-f011:**
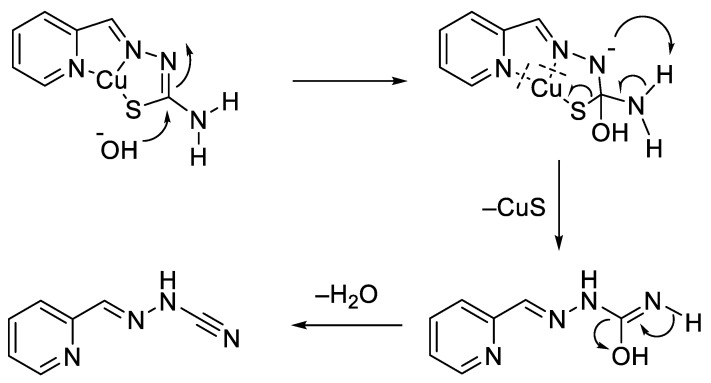
Mechanism of the base-catalyzed desulfurization of 3-AP/Cu complex, which generates the insoluble CuS and nitrile compounds.

**Figure 12 biomedicines-09-00852-f012:**
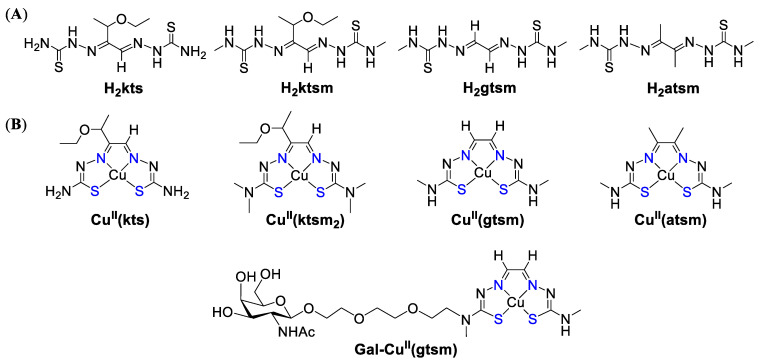
Chemical structures of (**A**) 2-keto-3-ethoxybutyraldehyde-bis(thiosemicarbazone) (H_2_kts), 2-keto-3-ethoxybutyraldehyde-bis(4-methylthiosemicarbazone) (H_2_ktsm), glyoxal-bis(4-methylthiosemicarbazone) (H_2_gtsm), diacetyl-bis(4-methylthiosemicarbazone) (H_2_atsm), (**B**) their respective Cu complexes and galactosamine (gal)-Cu(gtsm) complex.

## Data Availability

No new data were created in this study. Data sharing is not applicable to this article.

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
