# Peer review of "Modulation of Intracellular Copper Levels as the Mechanism of Action of Anticancer Copper Complexes: Clinical Relevance"

_biomedicines, 2021, doi:10.3390/biomedicines9080852_

Round 1
Reviewer 1 Report
This review has the potential to make a significant contribution to the 'copper in cancer' field. It provides a summary on the tested Cu-modifying compounds (i.e. Cu ionophores, Cu chelators, Cu-complexes), the bio-molecular mechanisms of action and their clinical potential.
But major adaptations need to be done before being considered for publications in Biomedicines journal.
Both review structure and text should be improved. More focus on the actual application of Cu modifying compounds in cancer disease is needed.
I suggest the authors to make a table listing the different tested compounds, with bio-molecular mechanism of action, and clinical application (results). This to support the reader in reading the review text.
From the abstract I understand that Cu-modifying compounds lead to higher cellular ROS and thereby cell death (oxidative stress response), but importantly this is not the only possible mechanism of action. Also mention other paths, t.ex. copper transport paths and specific cu binding proteins important in cell survival.
Some important references in the field of 'copper in cancer' should be added t.ex. reviews by Linda Vahdat, by Peter Friedl, and by Pernilla Wittung-Stafshede. More, you may consider adding Nick Tonks work on the Cu chelator DMP-1001 in Wilson´s disease, with potential application in cancer.
The authors should also enhance more how this review enriches the copper field, what conclusion can be made out of this summary on cu-modifying compounds and suggest how the copper field should evolve.
Author Response
The point-by-point reply is given in the attached document.

Reviewer 2 Report
attached as doc

Author Response
point-by-point reply is given in the attached document

Round 2
Reviewer 1 Report
Dear authors,
thank you for taking into consideration my earlier comments.
I have now only some Minor comments:
Line 174: adjust wording ’..2 times larger...’ ?
Line 196: I suggest to replace ‘negative growth regulation’ with different/more clarifying wording. From your description I understand that a Cu low diet leads to neoplasm formation, but your wording 'negative growth regulation' I understand the opposite is true, namely decrease in growth.
Line 203: ‘The increased acceleration’, replace by ‘the accelerated growth’
Line 207: ‘human umbilical artery and vein endothelial (HUVE) cells’ replace by ‘human umbilical vein endothelial cells (HUVEC)’
Line 229: ‘their proliferation rate was has markedly reduced’
Line 230: ´ …induced stimulatedion of maturation of undifferentiated progenitor cells…
Line 233: ‘treated with non-toxic concentration of CuCl2’ -> please specify the evaluated Cu concentration.
Line 254: replace ‘Cox’ by ‘COX’
In addition, please revise the text carefully for spelling and grammatical mistakes.
Reviewer 2 Report
The manuscript has been greatly improved and is acceptable in its present form